# OpenReview forum: "FreezeVLA: Action-Freezing Attacks against Vision-Language-Action Models"
_ICLR.cc/2026/Conference — ICLR 2026 Conference Withdrawn Submission_

### Official Review · Reviewer_aAWy · 2025-10-27

**Soundness:** 3
**Presentation:** 3
**Contribution:** 2
**Rating:** 4
**Confidence:** 4

**Summary:**

This paper proposes a minimax optimization procedure to adversarially attack the image embeddings in vision-language-action (VLA) models. In a similar spirit to the adversarial examples literature, the authors design a procedure that casts the task of finding adversarial images as a white-box two-play zero-sum game. The objective of the game is to maximize/minimize the probability (under a given VLA) of generating a particular <freeze> token.  The authors show the effectiveness of their attack across three different models on the LIBERO subsets.

**Overall assessment.** On the plus side, this paper considers a relatively new/under-explored problem and provides a full set of experiments.  On the negative side, the threat model is relatively contrived, there isn't anything particularly new about the algorithm (it's essentially PGD), there's a lack of comparisons to existing work (which is already cited in the paper), and there are missing details about the baselines. Overall, I tend to advocate for rejection when weighing these shortcomings against the positives. But I look forward to discussing this paper with the authors and the other reviewers.

**Strengths:**

- I agree that this is indeed an underexplored area of research. While a great deal of work has gone toward LLM-based jailbreaking, attacks on VLAs are much less studied. The authors are pushing in that direction, and there is novelty in doing so.
- The attack seems technically sound and principled. It inherits directly from PGD, although the authors include ablations that incorporate LLMs to generate seed prompts for the attack.
- The attack seems to work well at accomplishing the desired goal, which is to force the generation of the freeze token. The authors also do a number of ablations to understand when the attack is most effective.
- The figures in this paper are great!

**Weaknesses:**

- The baselines are not explained in enough detail. While the description in Table 1 is helpful to compare the baseline algorithms, it’s not clear what they actually do from the short paragraph starting on line 307.
- It’s not clear why a “fair” comparison results from this somewhat arbitrary choice of $\alpha$ and $\epsilon$. While it is true that these parameters are somewhat standard in the adversarial examples community (although $\epsilon=8/255$ is arguably much more standard than $\epsilon=4/255$), it’s not clear why an adversary should be confined to this range. The underlying assumption for restricting $\epsilon$ is generally that the attack should be imperceptible, but there’s no obvious reason why that should be the case. Moreover, it’s even more clear that there’s no reason to limit the step size of each attack; one could easily imagine it being the case that different step sizes are better for different algorithms.
- It’s not clear how the authors inserted the <freeze> token into the model’s vocabulary. It would seem not particularly straightforward to do this, since the action of freezing may correspond to several tokens (e.g., one for each degree of freedom).
- While the attack is designed for images, the goal of the attack seems somewhat similar to (Jones et al., 2025) in that the target is a particular action/set of actions. It would be worth comparing these two attacks, toward understand what is the most efficient way to generate a specific attack. Intuitively, the space of images seems much higher dimensional, and therefore may be more difficult to optimize in than text.
- I’d be curious to get the authors’ thoughts on the validity of the threat model. In what setting do we expect an adversary to _simultaneously_ have (a) white-box access to the target VLA, (b) access to 32 GPUS, and (c) the ability to apply arbitrary noise patterns during inference, which presumably happens at some nontrivial frequency (so as to properly actuate a robot in real time)? This seems relatively unlikely, especially when we compare to the somewhat more realistic threat model of textual jailbreaking (which the authors cite in the first few sections).

**Questions:**

- What does it mean to have, “black-box access to the user’s prompt?” To me, it seems like you can either know the prompt, or not know it. Generally, when constructing a threat model, the end-to-end system is viewed as white- or black-box. It’s somewhat different to say that you have _control_ over the user’s prompt, which is perhaps what is meant here.

---

### Official Review · Reviewer_2Tq1 · 2025-10-30

**Soundness:** 2
**Presentation:** 3
**Contribution:** 2
**Rating:** 2
**Confidence:** 4

**Summary:**

The paper introduces FreezeVLA, a two-stage adversarial attack targeting vision-language-action (VLA) models. Unlike prior work that causes robots to perform incorrect actions, this method aims to induce action-freezing: persistent inaction that could compromise safety in subtle ways. The authors design a min–max framework that first identifies “hard prompts” (instructions most resistant to freezing) and then optimizes adversarial images that freeze actions even under those prompts. Experiments on several open-source VLAs and the LIBERO benchmark demonstrate high attack success rates. However, this paper contains several issues that may face potential difficulties when it's applied.

**Strengths:**

1. It's interesting to see that the paper identifies “action-freezing” attacks as a distinct safety issue for VLAs. Most prior work focuses on causing wrong or dangerous actions; showing that “inaction” can itself be harmful is an interesting twist.

2. The min–max structure (first finding hard prompts, then optimizing an adversarial image) is clearly described.

3. The paper reports consistent attack success rates across multiple open-source VLAs and LIBERO splits. The performance gain from single-prompt to multi-prompt to the full FreezeVLA pipeline is clean.

**Weaknesses:**

**0. Discussion: A potentially narrow—but thought-provoking—formulation.**

**I also hope to see how other reviewers or the AC consider this formulation.**

At a conceptual level, FreezeVLA might be interpreted in two ways. On one hand, it highlights an interesting and underexplored attack objective: *persistent inaction*, which could reveal unique safety risks distinct from traditional misbehavior attacks. On the other hand, it could also be viewed as a special case of existing targeted-token attacks, where the target token simply corresponds to an inaction command (e.g., `<freeze>`).

Many previous adversarial settings already lead the robot to “freeze” or stall as a side effect of disrupted policy execution, which makes it unclear whether this behavior constitutes a fundamentally new attack category or merely a specific manifestation of known failure modes.

I personally find this boundary worth discussing: it may be narrow, but also conceptually interesting. I’d be curious to see how other reviewers or the AC interpret this distinction: whether FreezeVLA represents a genuinely new attack paradigm or simply reframes an existing one through the lens of inaction.

**1. Too close to existing multi-prompt or universal attacks in VLMs.**

The core pipeline (optimize over multiple prompts → aggregate gradients → universal perturbation) looks almost identical to prior VLM adversarial works like [1] Luo et al. (2024) and [2] Zou et al. (2023). The real change is that the “target” token is `<freeze>` instead of a generic wrong target, and the application scene is now in VLA. This feels incremental.

 **2. Questionable min-max design.**

One of the most confusing parts of the paper is the two-stage min–max design: first, the method searches for hard prompts (prompts that are difficult to “freeze”), and then it optimizes adversarial images that can paralyze the model even under those prompts.

Conceptually, this looks elegant on paper, but in practice it raises several issues:

**(a)** The inner loop performs gradient-based prompt editing (including synonym replacement) to push prompts toward “harder-to-freeze” directions. Then the outer loop optimizes a visual perturbation against those synthetic prompts. However, in the stated threat model, the attacker has *no control over or knowledge of the real user prompt*, and only the visual input can be modified. If that’s the case, why would the attacker spend significant compute searching for adversarial prompts first?

From an attack-efficiency standpoint, a direct universal image optimization across a representative prompt pool (without this inner prompt search) would make more sense, since you are conducting experiments on white-box models. The current two-stage pipeline feels more like a storytelling device than a necessity derived from the threat model.

**(b)** In Table 3, the so-called “hard prompts” quickly become grammatically broken or semantically nonsensical after a few optimization iterations — examples like “put the bowl on the see cable car” or “grasp towel from clean lemon table” show clear semantic drift. These aren’t realistic human instructions. Defeating such corrupted prompts doesn’t really demonstrate cross-prompt robustness; it just shows that the model is sensitive to meaningless input changes.

To me, this is a critical problem because the entire “hard prompt then attack image” story hinges on the assumption that those hard prompts represent genuinely challenging, but still valid, natural-language inputs. When the optimized prompts lose their human-plausible meaning, the evaluation loses external validity.

**(c)** Because the inner loop deliberately constructs highly adversarial or unnatural prompts, the final image attack only needs to work against those artificially difficult cases. It becomes a form of self-competition: the paper builds its own internal adversary (the corrupted prompts) and then reports that its attack can defeat it. This design can easily exaggerate performance numbers without proving true generality.

 **3. Simulation-only evaluation on an outdated benchmark and no real-world experiments.**

Every experiment is done on LIBERO, which was released years ago and is now widely overfitted.

Since “action-freezing” stability supposedly comes from a fixed viewpoint, it’s critical to test whether the attack still works when the camera or environment moves slightly. There’s no such test here. That makes the 90%+ ASR numbers hard to trust outside simulation.

The authors mention this under “Limitations,” but putting it there doesn’t excuse it. If the whole paper’s safety claim depends on purely simulated results, that’s a fundamental limitation, not a minor one. So, conducting more benchmarks and real-world experiments is a must.

 **4. No comparison with safety monitors or defenses.**

The paper claims that "This subtle yet stable inactivity can be easily mistaken for normal standby mode or successful task completion, enabling it to evade standard safety monitors”, yet doesn’t actually test any detection method (e.g., SAFE 2025 or failure monitors). This weakens the claim that it reveals a “critical safety risk.”


---

*References:*

[1] An image is worth 1000 lies: Transferability of adversarial images across prompts on vision-language models. ICLR 24.

[2] Universal and transferable adversarial attacks on aligned language models. 2023.

**Questions:**

Please see the weaknesses.

---

### Official Review · Reviewer_qWRT · 2025-10-30

**Soundness:** 2
**Presentation:** 3
**Contribution:** 2
**Rating:** 4
**Confidence:** 4

**Summary:**

This paper introduces FreezeVLA, a novel framework for creating adversarial attacks that “freeze” the actions of Vision-Language-Action (VLA) models by inducing action-stopping outputs through carefully crafted input image perturbations and prompt variants. The method combines robust prompt optimization with a min-max adversarial setup to generate inputs that consistently force the VLA model to produce a “freeze” action. The approach is tested on multiple representative VLA models.

**Strengths:**

1. The paper addresses action-freezing attacks against Vision-Language-Action (VLA) models, proposing a new adversarial problem that is both practically relevant and understudied in current literature.
2. The bi-level optimization approach, which jointly searches for robust prompts and adversarial image perturbations, is methodologically sound
3. The manuscript is well-organized and readable.

**Weaknesses:**

1. The description of the algorithm is confusing. It is unclear what variables such as k and m represent—are these epochs, iterative steps for hard prompt generation, or the number of reference prompts? The algorithm loops for each prompt, but clarification is needed. I recommend adding explanatory comments to Algorithm 1 or elaborating on the process in the main text, rather than simply stating “the procedure is outlined in Algorithm 1.”

2. The paper claims to target action-freezing attacks, but Table 1 only reports the overall success rate. It’s unclear how many cases actually resulted in the “freeze” action the method targets, which makes it difficult to evaluate the method’s claimed specificity and performance.

3. The paper would benefit from including visualizations of adversarial images generated by the proposed method to help readers qualitatively assess the attack.

4. I am skeptical about the effectiveness of the adversarial prompts generation procedure. As shown in Figure 5, intuitively, iterating more steps should produce prompts that are harder to defeat, making the model more robust. However, Figure 5 suggests that increasing the number of adversarial prompt steps has minimal impact on the attack success rate, especially compared to image perturbation steps.

5. Following previous point, another point is that the adversarial prompt maximization primarily substitutes single words with synonyms. This is a limited strategy compared to the broader scope of adversarial text attacks (e.g., paraphrasing, elaboration). The gradient-based word selection also tends to focus on action-related terms (as seen in Table 3), but failures in VLA models often stem from text-action or text-image misalignment. Changing normal words alone may not be sufficient; recent literature [1–4] shows substituting verbs or adjectives (such as “put”, “place”, “metal”) is more effective.

[1] Fang, Irving, et al. "From intention to execution: Probing the generalization boundaries of vision-language-action models." arXiv preprint arXiv:2506.09930 (2025).

[2] Li, Ao, et al. "Modeling Variants of Prompts for Vision-Language Models." arXiv preprint arXiv:2503.08229 (2025).

[3] Momeni, Liliane, et al. "Verbs in action: Improving verb understanding in video-language models." Proceedings of the IEEE/CVF International Conference on Computer Vision. 2023.

[4] Wu, Xiyang, et al. "On the Vulnerability of LLM/VLM-Controlled Robotics." arXiv preprint arXiv:2402.10340 (2024).

6. Perturbations are performed iteratively with variable K, but it’s unclear if the baselines used the same number of iterations for fairness. The number of perturbations can significantly influence attack success rates and should be explicitly controlled and reported.

**Questions:**

1. What are the computational requirements (time, resources) for the bi-level optimization process? How feasible is it to scale this approach to longer episodes or high-dimensional inputs?

2. Could you discuss or provide examples of failure cases where FreezeVLA is less effective, and what factors contribute to these limitations?

3. Have you tested FreezeVLA on tasks or models outside of your main experimental domain (e.g. real-world experiments)? How generalizable is the attack?

---

### Official Review · Reviewer_g73W · 2025-10-31

**Soundness:** 3
**Presentation:** 3
**Contribution:** 1
**Rating:** 2
**Confidence:** 4

**Summary:**

This paper introduces a new form of adversarial attack against vision-language-action (VLA) models. Unlike prior work which has focused on either a) task failure or b) adversarial trajectories, this work focuses on "action freezing" attacks, whereby the VLA stops in place and does not continue on its instructed trajectory. This paper contributes a new algorithm for this attack, that is able to achieve higher performance than the attacks it is compared to.

**Strengths:**

The paper is outlined clearly and provides well-written evidence to substantiate its claims. The proposed algorithm is a meaningful contribution to existing adversarial attack methods. The ablations and extra experiments provide depth which is useful in the context of the paper.

**Weaknesses:**

## Threat Model
The authors state that this work operates under a white-box threat model where the adversary can modify the image that gets passed into the VLA. I find this threat model incredibly difficult to motivate in the LIBERO environment. In particular, the adversarial image is overlaid on top of the reference image from the LIBERO environment. For this attack to be physically realizable, then, the adversary would have to have direct camera access in order to inject such a spacially-unaware adversarial image. A better threat model is found on page 7 of Wang et al.'s [paper](https://arxiv.org/pdf/2411.13587), where the authors demonstrate their attack in the real world with a sticker version of their adversarial patch.

Additionally, "action freezing" is the least dangerous proposed criterion for success of existing work. Other threat models, such as [task failure](https://arxiv.org/pdf/2411.13587) or [targeted arbitrary action sequences](https://arxiv.org/pdf/2506.03350) are significantly more dangerous than freezing. In fact, under adversarial pressure, freezing in place is perhaps one of the safest fallback actions for a VLA to take, as opposed to following an adversarially induced trajectory.



## Novelty
This paper lacks novelty compared to existing work. For example, [Jones et al.](https://arxiv.org/pdf/2506.03350) demonstrate an attack that is able to achieve any arbitrary target action on a VLA model, including the proposed `<freeze>` actions from this paper. Despite the work of Jones et al. focusing on white-box text access, the breadth of work evaluting VLAs against adversarial patches that already exists means that the proposed contribution of FreezeVLA is insufficiently significant in the context of current work.

**Questions:**

1. This attack relies on the concept of "reference prompts", which are sometimes generated by an LLM or randomly sampled (Table 1). However, section 3.2 states that they are always generated by an LLM. Can you please clarify this? For example, my interpretation of "reference prompts" with sampling would refer to the reference prompts for each LIBERO dataset, whereas w/ GPT refers to prompts generated by a LLM when given access to an image of the environment.
2. Relatedly, during the evaluation process, what prompts are you passing to the VLA? Are you passing user prompts or instructions that you have generated, or the specific prompts relevant to each LIBERO dataset?
3. Can you explain in better detail the reasoning behind choosing PGD and Multi-Prompt as baselines as opposed to UADA, UPA, TMA, and the variant of GCG from the Jones et al. paper linked earlier?
4. Would it be possible to provide some videos generated by the LIBERO evaluation pipeline to demonstrate the efficacy of the attack?
5. What is the specific token in the OpenVLA vocabulary that you have used for the `<freeze>` action? It is my understanding that every token that is produced gets evaluated by the action tokenizer into normalized continuous actions in the [0,1] action space for each degree of freedom. As a result, all 7 tokens that get output from the `predict_action` method will be in the [0,1] range. Under this, are you claiming that the `<freeze>` action is effectively `[0, 0, 0, 0, 0, 0, 0]`, or complete inaction across all 7 DoF?

---

### Note · Authors · 2025-11-24

I have read and agree with the venue's withdrawal policy on behalf of myself and my co-authors.